# Chrysanthemum Leaf Ethanol Extract Prevents Obesity and Metabolic Disease in Diet-Induced Obese Mice via Lipid Mobilization in White Adipose Tissue

**DOI:** 10.3390/nu11061347

**Published:** 2019-06-15

**Authors:** Ri Ryu, Eun-Young Kwon, Ji-Young Choi, Jong Cheol Shon, Kwang-Hyeon Liu, Myung-Sook Choi

**Affiliations:** 1Research Institute of Eco-friendly Livestock Science, Institute of Green-Bio Science and Technology, Seoul National University, Pyeongchang 25354, Korea; riryu@snu.ac.kr; 2Department of Food Science and Nutrition, Kyungpook National University, Daegu 41566, Korea; savage20@naver.com; 3Center for Food and Nutritional Genomics Research, Kyungpook National University, Daegu 41566, Korea; 4Lee Gil Ya Cancer and Diabetes Institute, Gachon University, Incheon 21936, Korea; jyjy31@hanmail.net; 5Enviromental Chemistry Research Center, Korea Institute of Toxicology, Jinju 52384, Korea; jongcheol.shon@kitox.re.kr; 6BK21 Plus KNU Multi-Omics Based Creative Drug Research Team, College of Pharmacy and Research Institute of Pharmaceutical Sciences, Kyungpook National University, Daegu 41566, Korea; dstlkh@knu.ac.kr

**Keywords:** *Chrysanthemum morifolium* Ramat, obesity, lipid mobilization, energy expenditure, transcriptome

## Abstract

This study aimed to elucidate the molecular mechanism of *Chrysanthemum morifolium* Ramat. against obesity and diabetes, by comparing the transcriptional changes in epididymal white adipose tissue (eWAT) with those of the bioactive compound in *C. morifolium*, luteolin (LU). Male C57BL/6J mice were fed a normal diet, high-fat diet (HFD), and HFD supplemented with 1.5% w/w chrysanthemum leaf ethanol extract (CLE) for 16 weeks. Supplementation with CLE and LU significantly decreased the body weight gain and eWAT weight by stimulating mRNA expressions for thermogenesis and energy expenditure in eWAT via lipid mobilization, which may be linked to the attenuation of dyslipidemia. Furthermore, CLE and LU increased uncoupling protein-1 protein expression in brown adipose tissue, leading to energy expenditure. Of note, CLE and LU supplements enhanced the balance between lipid storage and mobilization in white adipose tissue (WAT), in turn, inhibiting adipocyte inflammation and lipotoxicity of peripheral tissues. Moreover, CLE and LU attenuated hepatic steatosis by suppressing hepatic lipogenesis, thereby ameliorating insulin resistance and dyslipidemia. Our data suggest that CLE helps inhibit obesity and its comorbidities via the complex interplay between liver and WAT in diet-induced obese mice.

## 1. Introduction

Since obesity is associated with an increased risk of metabolic disorders, including insulin resistance, type 2 diabetes, dyslipidemia, fatty liver, hypertension, and various chronic diseases, it is recognized as a growing health problem. White adipose tissue (WAT) serves an important role as a surplus energy storage, and also has an endocrine function, synthesizing biologically active compounds and hormones that regulate metabolic homeostasis [1]. In the obese state, expansion of the impaired adipocytes increases the secretion of pro-inflammatory adipokines, such as leptin, resistin, tumor necrosis factor-α (TNF-α), interleukin (IL)-6, IL-18, macrophage inflammatory protein-1β (MIP-1β), and monocyte chemotactic protein 1 (MCP-1), which can lead to inflamed adipose tissue, macrophage accumulation, and the synthesis of fibrotic components [2]. In this way, dysfunctional adipose tissue drives the development of chronic low-grade inflammation, insulin resistance, and obesity-associated complications [3,4,5]. In addition, obesity and high-fat diet (HFD) feeding also promote lipolysis and decrease lipogenesis, in turn, perturbing cellular insulin signaling [6,7,8]. Decreased storing capacity of adipose tissue may contribute to elevated circulating plasma free fatty acids (FFAs) and ectopic fat deposition via the portal system, which causes lipotoxicity of non-adipose tissues, including liver, skeletal muscle, kidney, and pancreas [9]. Furthermore, accumulated ectopic lipid in the liver leads to hepatic steatosis via overexpression of transcription factors, such as the sterol regulatory element-binding proteins [10], resulting in over-secretion of very low-density lipoprotein (VLDL) and downstream effects on atherogenic dyslipidemia [11].

Due to the global epidemic of obesity, extensive research is focused on its prevention, diagnosis, and treatment. Recent approaches to targeting energy expenditure and thermogenesis have emerged in the use of drugs or functional food to counteract obesity treatment [12]. Natural products possess multi-compound and multi-target characteristics, and so are likely to exhibit synergistic effects. Multi-target drugs have attracted considerable attention, due to their advantages in the treatment of complex diseases and minimal undesirable side effects [13].

The flower of chrysanthemum (*Chrysanthemum morifolium* Ramat.) is popular worldwide and has demonstrated potential effects against insulin resistance, inflammation, and cardiovascular disease [14,15] that have been associated with its flavonoids, such as luteolin (LU), chlorogenic acid, apigenin, and acacetin [16]. Research shows that LU supplements exert protective effects against obesity and its comorbidities via interplay between the liver and adipose tissue in diet-induced obese mice [17]. However, its molecular mechanism has not yet been clarified. Therefore, the current study applied transcriptomic analysis using RNA sequencing (RNA-Seq), to elucidate how chrysanthemum leaf modulates metabolic regulation, and decipher the mechanism underlying the multi-target efficacy of its major component, LU, in C57BL/6J mice fed an HFD.

## 2. Material and Methods

### 2.1. Preparation of Chrysanthemum morifolium Ramat. Ethanol Extract (CLE)

CLE was obtained from Haihang Industry Co., Ltd. (Jinan, China). The raw material, *C. morifolium* Ramat. was extracted three times with 10 volumes of 70% ethanol in an extraction pot at 65–75 °C for 2 h per extraction. The CLE was collected, concentrated extraction liquid, and then spray-dried (120 °C, 80 mesh). The yield was 10%. Flavonoids in CLE were identified using liquid chromatography–tandem mass spectrometry and are listed in Appendix A.

### 2.2. Experimental Design

Four-week-old male C57BL/6J mice were obtained from Jackson Laboratory (Bar Harbor, ME, USA). All mice were individually housed under constant temperature (24 °C) and 12-h light/dark cycle. After a 1-week acclimation on a normal chow diet, the mice were then randomly divided into four groups, and each group was fed one of the following diets for 16 weeks: Normal diet (ND, 12% kcal from fat, *n* = 10); HFD (45% of kcal from fat, *n* = 15); HFD with 1.5% (*w*/*w*) CLE (*n* = 10); HFD with 0.003% (*w*/*w*) LU (*n* = 10) (Appendix A). The dose of CLE and luteolin was based on previous in vivo studies, 0.005% (*w*/*w*) luteolin and 5% (*w*/*w*) chrysanthemum flower extract, respectively [15,17]. 0.003% (*w*/*w*) luteolin and 1.5% (*w*/*w*) CLE were decided for in investigating the biological effects at the lower dose than doses in the previous study. At the end of the experimental period, all mice were anesthetized with isoflurane after 12-h of fasting. Animal studies were performed under the protocols approved by the Kyungpook National University Industry Foundation (Approval No. KNU-2016-49).

### 2.3. Measurement of Calorimetry and Oxygen Consumption

Energy expenditure (EE) was measured using an indirect calorimeter (Oxylet; Panlab, Cornella, Spain). The mice were placed into individual metabolic chambers at 25 °C, with free access to food and water. O_2_ and CO_2_ analyzers were calibrated with highly purified gas standards. Whole-body oxygen consumption (VO_2_) and carbon dioxide production (VCO_2_) were recorded at 3-min intervals using a computer-assisted data acquisition program (Chart 5.2; ADInstruments, Sydney, Australia) over a 24-h period, and the data were averaged for each mouse. The respiratory quotient (RQ) is automatically calculated as the ratio VCO_2_ and VO_2_.The EE was calculated as follows:EE (kcal/day/kg of body weight) = VO_2_ × 1.44 × [3.815 + (1.232 × VO_2_/VCO_2_)](1)

### 2.4. Plasma, Hepatic, and WAT Lipid Contents

The plasma triglycerides (TGs), total cholesterol (total-C), and high-density lipoprotein cholesterol (HDL-C) were determined using commercial enzymatic kits (Asan Pharmaceutical Co., Seoul, Republic of Korea). Enzymatic kits used to analyze apolipoprotein AI (Apo AI) and Apo B levels were from Eiken Chemical (Tokyo, Japan) while those used to measure the plasma FFA and phospholipid (PL) levels were supplied by Wako Chemicals (Richmond, VA, USA). Non-HDL-cholesterol (non-HDL-C) and the HDL-C-to-total-C ratio (HTR) were calculated using Equations (2) and (3), respectively:Non-HDL-C = [total-C (mmol/L) − HDL-C (mmol/L)](2)
HTR = [(HDL-C/total-C × 100)](3)

Hepatic and epididymal white adipose tissue (eWAT) lipids were extracted (Folch et al., 1957 [18]), and then the dried lipid residues were dissolved in 1 mL ethanol for TG and cholesterol assays. Triton X-100 and a sodium cholate solution in distilled water were added to 200 μL of the dissolved lipid solution for emulsification. The TG, total-C, and FFA concentrations were assayed using the same enzymatic kit used for the plasma analyses above.

### 2.5. Plasma Biochemical Profiles, Fasting Blood Glucose, Homeostatic Model Assessment of Insulin Resistance (HOMA-IR), and Hepatic Glycogen

The HOMA-IR index was calculated according to the homeostasis of the assessment as shown below:HOMA-IR = [fasting glucose (mmol/L) × fasting insulin (μL U/mL)]/22.5(4)

Plasma hormones (insulin and C-peptide), adipokines (leptin, adiponectin, and resistin), and plasma cytokines (MCP-1 and MIP-1β) were evaluated using a multiplex detection kit (Bio-Rad, Hercules, CA, USA).

### 2.6. Plasma Glutamic Oxalacetic Transaminase (GOT) and Glutamic Pyruvic Transaminase (GPT) Activities Measurement

The GOT and GPT activities were measured using a commercially available kit (Asan Pharmaceutical Co.).

### 2.7. Histological and Immunohistochemistry Analyses

Liver and eWAT were removed from the mice and fixed in a buffer solution of 10% formalin. All fixed tissues were processed routinely for paraffin embedding, and 4-μm-thick sections were prepared and stained with hematoxylin and eosin (H&E) and Masson’s trichrome stain. For immunohistochemistry, a polyclonal anti-uncoupling protein-1 (UCP1) antibody (Abcam, UK) was diluted 1:50. The stained areas were viewed using an optical microscope (Nikon, Tokyo, Japan) with a magnifying power of ×200.

### 2.8. mRNA-Seq and Molecular Network Analysis

Total RNA in eWAT was extracted from three representative samples of each group using TRIzol reagent (Invitrogen Life Technologies, Grand Island, NY, USA), according to the manufacturer’s instructions. The libraries of mRNA-Seq were prepared as paired-end reads with a length of 100 bases, using a TruSeq RNA sample preparation kit (Illumina, San Diego, CA, USA). mRNA molecules were purified and fragmented from 2 μg total RNA, using oligo(dT) magnetic beads. The fragmented mRNAs were synthesized as single-stranded cDNA, using random hexamer priming. By applying this as a template for second strand synthesis, double-stranded cDNA was prepared. After the sequential processes of end-repair, A-tailing, and adapter ligation, cDNA libraries were amplified by PCR. The quality of these cDNA libraries was evaluated by the Agilent 2100 bioanalyzer (Agilent, Santa Clara, CA, USA). cDNA libraries were quantified using the KAPA library quantification kit (Kapa Biosystems, Wilmington, MA, USA), according to the manufacturer’s library quantification protocol. After cluster amplification of denatured templates, flow cells were sequenced as paired-ends (2 × 100 bp) using an Illumina HiSeq2500 machine. For differential expression analysis, gene-level count data were generated using Cuffdiff software v2.2.1. Based on the calculated read count data, differentially expressed genes (DEG) were identified using the DEGseq R package. DEGseq supports users to export gene expression values in a table format that can be directly processed by edgeR [19]. DEGseq can also be applied to identify differential expression of exons or pieces of transcripts [20]. Normalization factors were calculated using the iterative DEGES/edgeR method. The *q*-value was calculated based on the *p*-value, using the p.adjust function of the R package, with default parameter settings. DEG were identified based on *q*<0.05. Functional annotation clustering of the gene ontology (GO) terms was analyzed by using Database for Annotation, Visualization and Integrated Discovery (DAVID), wherein a higher enrichment score signifies more cluster enrichment. The genes are listed in Appendix A. Furthermore, the molecular network of DEG was analyzed using Ingenuity Pathway Analysis (IPA, Ingenuity^®^ Systems, Qiagen, Redwood City, CA, USA). The RNA-seq data has been submitted to the publicly available NCBI’s Gene Expression Omnibus Database (http://www.ncbi.nlm.nih.gov/geo/): accession number GSE 130218.

### 2.9. Gene Expression Analysis

Total RNA (1 μg) was reverse-transcribed into cDNA using the QuantiTect^®^ reverse transcription kit (Qiagen, Hilden, Germany). mRNA expression was then quantified by real-time quantitative PCR (RT-qPCR) using the SYBR green PCR kit (Qiagen, Germany) and the CFX96TM real-time system (Bio-Rad). Gene-specific mouse primers were used, as indicated in Appendix A. The amplification was performed as follows: 90 °C for 10 min, followed by 35 cycles of 95 °C for 15 s and 60 °C for 60 s. The cycle threshold (*Ct*) levels were normalized to 36B4, and the relative gene expression was calculated using the 2^−ΔΔ^*^Ct^* method [21].

### 2.10. Statistical Analysis

Data were expressed as the mean ± standard error of the mean. All statistical analyses were performed using the SPSS for Windows software v23.0 (SPSS, Inc., Chicago, IL, USA). Statistical significance between the ND and HFD groups were determined using the Student’s *t*-test. In addition, statistical significances among HFD feeding groups were determined using one-way analysis of variance (ANOVA), with post hoc analysis by Duncan’s multiple-range in the SPSS software. Differences were considered statistically significant at *p* < 0.05.

## 3. Results

### 3.1. CLE Supplement Suppresses Increase in Body Weight Gain and Fat Deposition

The body weight of ND mice was significantly reduced compared with mice fed the HFD. Both CLE and LU supplements significantly decreased the body weight from week one, relative to the HFD-fed mice (Figure 1A). Among the mice fed the experimental diets, the daily body weight gains over 16 weeks were significantly lower in the CLE and LU groups. The food efficiency ratio (FER) was significantly decreased in both the CLE and LU groups compared with the HFD group (Figure 1B).

In addition, the weights of organs (liver, kidney, and muscle) and adipose tissue were measured and expressed per 100 g body weight. Supplementation of CLE and LU significantly increased the liver weight, and the muscle weight also increased compared with the HFD group (Figure 1C). In comparison to mice fed the HFD, CLE supplementation significantly lowered the epididymal, perirenal, mesenteric, visceral, subcutaneous, and total WAT, as well as the interscapular brown adipose tissue (BAT) mass (Figure 1D).

### 3.2. CLE Supplement Regulates Lipid Metabolism and EE

Both the CLE and LU groups exhibited a smaller epididymal adipocyte size than the HFD group, consistent with the adipose tissue mass (Figure 2A). Supplementation of CLE and LU significantly increased EE and VO_2_ during a 12-h light/dark cycle when compared with the HFD group (Figure 2B). The ND group exhibited a high RQ compared to the HFD-fed groups. RQ was lower during the dark cycle in CLE and LU groups than in the HFD group, suggesting that CLE might utilize carbohydrate in HFD-fed mice. Notably, the TG concentration per gram of eWAT was significantly lowered in the HFD group than in the ND group, the CLE and LU supplement did not affect it. However, when the TG concentration converted in total eWAT, the result is reversed by showing markedly elevated compared to the ND group due to increased weight of eWAT in HFD group. In addition, fatty acid (FA) concentration was significantly higher in the HFD group than in the ND group. However, CLE and LU supplements significantly reduced both the concentrations of FA/g and FA in total eWAT than HFD-fed mice (Figure 2C), consistent with reduced plasma FFA level. Therefore, we evaluated the expression of the thermogenic and fatty acid oxidation (FAO) genes by qPCR analysis in eWAT. As a result, the CLE supplement significantly upregulated the *Ucp1*, *Cox8b*, *Cidea*, *fgf21*, and *Adrb3* genes (Figure 2D). Along with thermogenic genes, the FAO genes (*Cpt2*, *Acox1*, *and Acadl*) were increased by CLE, together with increased expression of their transcription regulators (*Ppara*, *Ppargc1a* (PGC1α), and *Ppargc1b* (PGC1β)). In BAT, immunohistochemistry analysis showed a significant increase in UCP1 protein expression, as well as smaller BAT adipocyte size in mice supplemented with CLE compared with the HFD group.

### 3.3. CLE Supplement Prevents Dyslipidemia

Mice fed the HFD exhibited a significant increase in plasma lipid levels, except plasma HDL-C and Apo A1 levels. However, supplementation of CLE and LU significantly lowered the plasma PL, total-C, non-HDL-C, and Apo B levels compared with the HFD group, whereas plasma HDL-C and Apo A1 levels were significantly higher in the CLE group (Table 1). Plasma FFA level was significantly decreased in the CLE group by the Student’s *t*-test. Furthermore, the CLE group had a significantly lower HTR than the HFD group.

### 3.4. CLE Supplement Suppresses Hepatic Steatosis and Hepatic Toxicity

Since the CLE and LU groups exhibited an improvement in plasma lipid levels, hepatic steatosis was observed by H&E staining, and hepatic lipid concentrations, such as TGs, FAs, and cholesterol, were measured and expressed per gram of liver. Both groups revealed visible morphological evidence of a reduction in the number and size of the lipid droplets (LDs), which is indicative of increased hepatic fat accumulation and LDs in HFD-fed mice (Figure 3A). The concentrations of TGs, FAs, and cholesterol were significantly lowered in the CLE group. In addition, the hepatic TG and cholesterol levels were significantly lower in the LU group compared with the HFD group (Figure 3B). Consistent with the hepatic lipid levels, the mRNA expression of lipid metabolism-related genes was evaluated in the liver (Figure 3C). CLE significantly downregulated the expression of the lipogenic genes, such as *Pparg*, *Srebf1a*, *Srebf1c*, *Srebf2*, *Fasn*, *scd1*, and *Acaca* (ACC1) in comparison to the HFD group. The expression of the FAO gene *Cpt1a* was increased in both the CLE and LU groups relative to the HFD group. Moreover, supplementation of CLE and LU significantly decreased the hepatotoxicity markers, namely the GOT and GPT levels, relative to the levels in the HFD group (Figure 3D).

### 3.5. CLE Supplement Improves Insulin Resistance

The plasma glucose and insulin levels were significantly reduced by CLE and LU supplements compared with the HFD group. CLE also decreased plasma C-peptide, a marker of insulin secretion, and both CLE and LU supplements significantly lowered the HOMA-IR index, indicating an improvement of insulin resistance (Figure 4).

### 3.6. CLE Supplement Reduces Inflammatory Markers

Supplementation of CLE led to significant reductions in plasma leptin, resistin, MIP-1β, and MCP-1 (also known as CLL2) levels. The plasma adiponectin level was similar between the CLE group and HFD group, but significantly higher in the LU group than the HFD group (Figure 5A). However, the leptin/adiponectin ratio, which is a marker of metabolic disease and obesity, was decreased in both the CLE and LU groups. In addition, the gene expression of *Adipoq* (adiponectin), an anti-inflammatory gene, was upregulated, whereas *Tnfa*, *Tlr2*, and *Tlr4* were downregulated in both the CLE-and LU-supplemented groups compared to the HFD control group (Figure 5B).

### 3.7. CLE Supplement Alters the Transcriptional Response in eWAT

To determine the molecular basis of the phenotype of CLE and LU supplementation, the transcriptomic profile of the adipocytes derived from eWAT were investigated by the mRNA-Seq analysis. The DEG were identified by the following cut-off values: Fold change > 1.5; *q* < 0.05 (Figure 6A). In the HFD group, 2501 DEG were upregulated, and 1717 DEG were downregulated compared with the ND group, in the eWAT. Meanwhile, the CLE supplement upregulated 1076 genes and downregulated 1578 genes compared with the HFD group. A total of 759 DEGs were upregulated, and 1304 DEGs were downregulated in the LU group relative to the HFD group. In addition, a total of 1823 common genes were identified among the CLE and LU groups (Figure 6B). However, the CLE supplement only 831 DEGs were regulated, and the LU supplement only 240 DEGs were upregulated in among the high-fat diet-responsive genes (Figure 6B). Functional annotation clustering of the GO terms in response to CLE and LU was performed using the DAVID program (Figure 6C,D). The 1076 genes upregulated by CLE supplementation were enriched in biological processes related to the insulin signaling pathway, diet-induced thermogenesis, positive regulation of glucose import, and positive regulation of FA β-oxidation, and stimulation of glycoprotein (Figure 6C). Conversely, the 1578 genes downregulated by CLE supplementation, the enriched GO terms were involved in the immune system process, lysosome, collagen, proteolysis, and the toll-like receptor signaling pathway. Similarly, the GO analysis of the 759 genes upregulated by LU supplementation revealed enrichment of glycoprotein, the glutathione metabolic process, diet-induced thermogenesis, and the cellular response to bone morphogenetic protein (Figure 6D). Among the 1304 genes downregulated by LU supplementation, the immune system process, lysosome, collagen, the lipid metabolic process, and proteolysis were enriched GO terms. Moreover, the CLE and LU groups have common GO terms, which is thermogenesis, lipid metabolism and immune system process, while insulin signaling GO term is enriched by CLE supplement.

Genes that were differentially expressed were analyzed using the IPA software. When overlay analysis was performed, lipolysis was activated in the HFD group when compared with the ND group; however, CLE inhibited excess lipolysis compared with the HFD group (Figure 7A). Moreover, of the biofunctions related to obesity, both inflammation and abnormality of adipose tissue were inhibited by CLE supplementation (Figure 7B).

## 4. Discussion

In the present study, HFD-fed mice reduced the TG/g content in eWAT and increased the FA concentration in both plasma and eWAT compared with ND-fed mice, indicating HFD induced and regulated adipose lipolysis. HFD downregulated *Pparg* (PPARγ), a master regulator of the lipogenesis-related gene, in eWAT compared with the ND group. Whereas, the HFD group showed a decreased expression of FAO-related genes in eWAT. We also discovered that HFD feeding leads to dysfunctional lipid metabolism by downregulation of lipogenesis and FAO, along with upregulation of lipolysis. In contrast, CLE supplementation decreased the FA level in eWAT, but the TG level was not altered. FAO-related genes (*Pparα*, *Ppargc1a*, *Ppargc1b*, *Cpt2*, *Acox1*, and *Acadl*) were downregulated, whereas the expression of *Pck1* and *Pparg* genes were upregulated by CLE supplementation, in eWAT. In line with these data, IPA analysis from mRNA-Seq data of eWAT revealed that CLE inhibited obesity-related biofunction and abnormality of adipose tissue. Moreover, CLE supplementation inhibited excess lipolysis of adipose tissue activated by HFD feeding. Activated PPARγ stimulates the expression of genes involved in FA uptake in WAT [22]. In CLE-supplemented mice, adipocyte FA uptake gene (*Ffar4* and *Fabp1*) expressions were increased in eWAT, whereas HFD feeding decreased their expression. Ffar4, known as GPR120, promotes adipogenesis via PPARγ but also attenuates low-grade inflammation and insulin resistance accompanying obesity [23,24]. In the transcriptomic analysis, even if CLE relatively attenuated lipolysis of adipose tissue compared with the HFD group, major regulators, namely *Plin1* and *Pnpla2* transcripts, were upregulated in eWAT. Moreover, mRNA expression of *Plin2* (Adrp), which is associated with the formation of LDs, was downregulated in response to CLE. These transcriptional regulations might contribute to reducing the size and number of adipocytes in eWAT, as seen in the morphological observation. Thus, these results indicate that increased *Pparg* promotes FA uptake, and those FAs are oxidized via FAO-related genes, resulting in a significant decrease in plasma FFA level by CLE supplementation. Conversely, HFD failed to activate FA oxidation, causing an increase in the FA level in both plasma and WAT, which, in turn, protects against lipotoxicity, as demonstrated by lowered levels of hepatotoxicity markers, such as plasma GOT and GPT levels. In agreement with a previous study by our group, LU could prevent hepatic lipotoxicity through the decrease in FA flux to the liver by increasing the expression of FA uptake and adipogenic genes, leading to FA re-esterification [17], and suggesting a lipid-lowering action by CLE and LU.

In this study, we discovered that CLE and LU supplements could improve insulin resistance. Increased flux of FFA level by enhancement of adipose lipolysis mainly leads to TG accumulation via FA esterification in the liver, and then newly synthesized TGs are restored as LDs and secreted as VLDL during obesity and insulin resistance because FFAs also stimulate insulin secretion [25,26]. Therefore, both hyperglycemia and hyperinsulinemia promote hepatic *de novo* lipogenesis and reduce β-oxidation, thereby leading to hepatic steatosis and glucose output [26]. In this study, HFD elevated hepatic lipid profiles; however, CLE significantly decreased hepatic TGs, FAs, and cholesterol with a reduction of lipid accumulation in the liver. In the liver, CLE supplementation decreased the expression of lipogenic genes, such as *Acaca* (*ACC1*), *Fasn*, *Scd1*, *Srebf1a*, *Srebf1c*, *Srebf2*, and *Pparg* while it increased the FAO-related gene, *Cpt1a*. Of note, PPARγ promotes hepatic steatosis while decreased PPARγ function protects against hepatic steatosis in diet-induced obesity and insulin resistance mice [27,28]. In contrast, mice without PPARγ in the adipose tissue develop TG accumulation and insulin resistance [29]. Thus, CLE improves hepatic steatosis by reducing *Pparg* gene expression while simultaneously enhancing *Pparg* gene expression in adipose tissue, which leads to preventing insulin resistance. Recent studies suggested that LU enhanced insulin sensitivity in diet-induced obese mice [17,30] and 3T3L-1 adipocytes by activating the PPARγ pathway [31]. In addition, Yamamoto et al. (2015) reported that supplementation of the hot water extract of chrysanthemum flower (5%, w/w), which also contains LU, increased gene and protein expressions of glucose transporter type 4 and PPARγ in eWAT of diabetic KK-A^y^ mice. Consistent with mRNA expressions related to hepatic lipogenesis, morphological characterization evidenced a reduction in hepatic LDs, suggesting dietary CLE attenuated hepatic steatosis through limiting lipogenesis in the liver. Since hepatic steatosis is recognized as a primary contributor to systemic insulin resistance, CLE may improve insulin sensitivity by suppressing the development of hepatic steatosis. Indeed, plasma glucose concentration and HOMA-IR, a marker of insulin resistance, were also improved in both the CLE and LU groups. Likewise, plasma C-peptide, which reflects insulin secretion, was decreased in CLE- and LU-supplemented mice, accompanied by a significant decrease in plasma insulin level. In line with these results, the transcriptional pattern revealed that CLE supplementation upregulated glucose metabolism-related genes.

As adiposity increases, secretion of inflammatory cytokines from WAT is promoted, whereas, the secretion of anti-inflammatory adipokines is decreased. Importantly, chronic inflammation of adipose tissues could also induce inflammation in metabolic organs, including liver and muscle [32]. Thus, obesity is associated with chronic low-grade inflammation, which can cause the development of type 2 diabetes and obesity-related comorbidities [3,4,5]. With HFD feeding in mice, plasma MCP-1 and MIP-1β levels were elevated, but CLE and LU supplements lowered these plasma levels and reduced body weight. Plasma leptin and resistin levels were elevated, and adiponectin level was decreased in mice fed the HFD, whereas CLE supplementation decreased plasma leptin and resistin levels. Additionally, the leptin:adiponectin ratio, which is used as an index of insulin resistance along with HOMA-IR, was reduced in the CLE group. This trend accounts for the decreased plasma leptin concentration, despite no difference in plasma adiponectin level. In addition, transcriptomic analysis of the eWAT revealed that CLE downregulated *lep* and *Tnf* mRNA expression, and upregulated adiponectin and genes encoding adiponectin receptors, such as *Adipoq* (adiponectin), *Adipor* (adiponectin receptor) *1* and *Adipor2*. Adiponectin receptors are thought to play key roles in metabolic syndrome and life span [33]. LU supplementation not only increased *Adipor2* gene expression and plasma adiponectin level but also decreased plasma leptin, resistin, the leptin:adiponectin ratio, and the *Tnf* mRNA level. Together, it is plausible that the beneficial effect on the improved inflammatory markers and metabolic regulation by the CLE supplement is due to the enhanced expression of *Adipoq* and *Adipors* genes. Moreover, pro-inflammatory cytokines stimulate lipolysis in adipocytes, which, in turn, causes lipotoxicity of peripheral tissues [32]. In enlarged adipocytes from chronically inflamed adipose tissue, TNF-α and its receptors can promote lipolysis and secretion of FFAs in adipose tissue via inhibiting esterification of FAs to TGs [7,34]. Notably, elevated TNF-α diminished the function of the adipogenic regulator, PPARγ. Since PPARγ has not only a modulatory role in adipogenesis or metabolism but also inhibitory effects on inflammation by reducing inflammatory molecules or genes in macrophages because it negatively interferes with the signaling pathways of nuclear factor κB, signal transducer and activator of transcription, and activating protein-1 [35,36]. PPARγ down-regulation could induce aggravation of adipocyte inflammation, in turn resulting in obesity-regulated disease. Consistent with this notion, HFD enhanced lipolysis in adipose tissue accompanied by upregulation of the *Tnf* gene and downregulation of the *Pparg* gene in eWAT. However, supplementation of CLE and LU reversed these markers in eWAT. These results suggest that the CLE supplement has an improvement effect on inflammation of adipose tissue.

In the present study, the CLE supplement changed biochemical profiles and gene expression pattern in similar with the LU group. However, CLE/HFD regulated a various gene expression more than LU/HFD in mRNA sequencing analysis, although a total of 1823 common genes were identified among the CLE and LU groups. CLE contained luteolin 0.008%, only 4 % of CLE. It is supposed that CLE has synergy effect in combination luteolin, its glycoside and chlorogenic acid. Among flavonoids present in CLE, luteolin-7-O-β-D-glucuronide exhibited antioxidative and AGE-inhibitory activities [37]. Chlorogenic acid is well known to have anti-obesity and anti-diabetic effects [38]. This suggests a synergistic effect of the bioactive phytochemical compounds in CLE, but luteolin seemed to be more potent for obesity and related metabolic disorders than other compounds. Thus, these results indicate that luteolin in CLE may be responsible for its bioactive compound.

## 5. Conclusions

In conclusion, the present study demonstrated that CLE and LU supplements might improve obesity and its comorbidities, including insulin resistance, hepatic steatosis, and inflammation, by regulating glucose and lipid metabolism in HFD-induced obese mice. CLE and LU exhibited beneficial effects on obesity and dyslipidemia by increasing gene and transcript expressions for thermogenesis via lipid mobilization in eWAT, which, in turn, increases EE. At the same time, CLE down-regulated *Tnf*, which led to upregulation of *Pparg*, a regulator of lipogenesis and FA uptake genes in eWAT, which is linked to an attenuation of FA flux into other peripheral tissues, such as the liver, thereby alleviating lipotoxicity. In contrast to eWAT, CLE and LU suppressed the expression of lipogenic genes in the liver, ameliorating hepatic steatosis and, consequently, enhancing hepatic insulin sensitivity and dyslipidemia. Moreover, enhanced lipid storage capacity in adipose tissue reduced inflammation.

## Figures and Tables

**Figure 1 nutrients-11-01347-f001:**
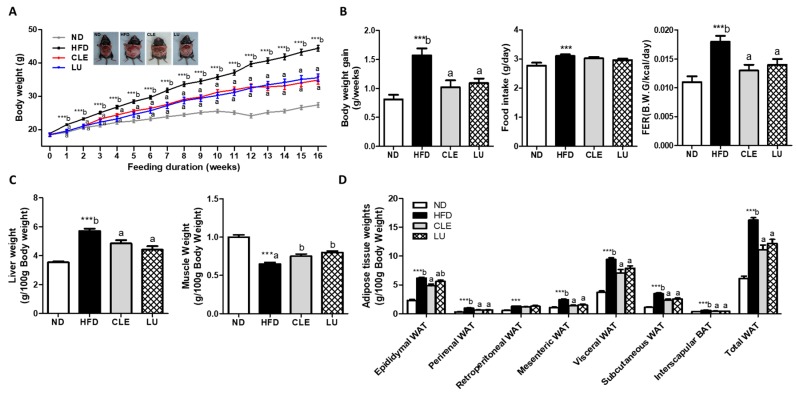
Effect of chrysanthemum leaf ethanol extract and luteolin supplements for 16 weeks on body weight in C57BL/6J mice fed a high-fat diet. (**A**) Weekly changes in body weight; (**B**) body weight gain, food intake, and food efficiency ratio (FER, body weight gain/energy intake); (**C**) organ weights (**D**) adipose tissue weights. Data are mean ± SE. Significant differences between HFD compared to ND are indicated *** *p* < 0.001 (Student’s *t*-test). ^ab^ Means not sharing a common letter are significantly different among the HFD-fed groups at *p* < 0.05. ND, normal diet (AIN-76); HFD, high-fat diet (45% kcal from fat); CLE, HFD+chrysanthemum leaf ethanol extract (1.5%, *w*/*w*); LU, HFD+luteolin (0.003% *w*/*w*). WAT, white adipose tissue; BAT, brown adipose tissue.

**Figure 2 nutrients-11-01347-f002:**
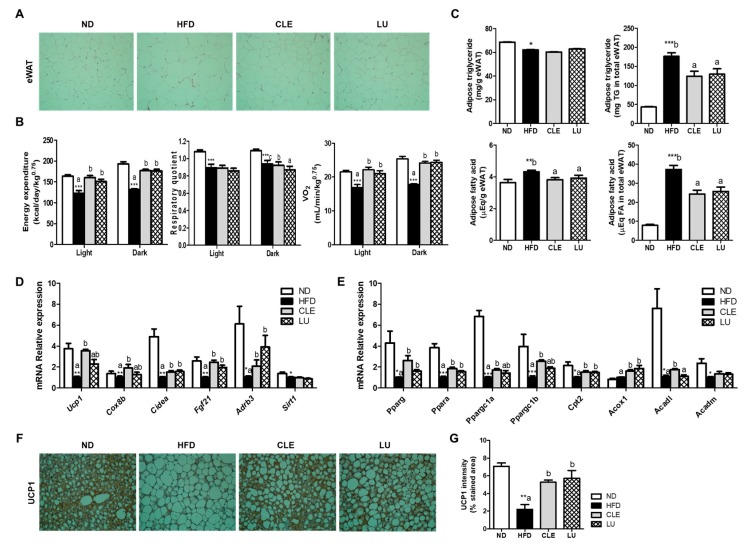
Effect of chrysanthemum leaf ethanol extract and luteolin supplements for 16 weeks on anti-adiposity. (**A**) eWAT morphology (magnification ×200); (**B**) energy expenditure, respiratory quotient and oxygen consumption (VO_2_); (**C**) lipid profile of eWAT; (**D**) adipocyte gene expression of thermogenic genes; (**E**) adipocyte gene expression of lipid metabolism-related genes; (**F**) immunohistochemistry of UCP1 staining in BAT (magnification ×200); (**G**) intensity of UCP1 in BAT. Mean ± SE. Significant differences between HFD compared to ND are indicated * *p* < 0.05, ** *p* < 0.01, *** *p* < 0.001 (Student’s *t*-test). ^ab^ Means not sharing a common letter are significantly different among the HFD-fed groups at *p* < 0.05. Hematoxylin and eosin- and UCP1 immunohistochemistry-stained transverse-section of epididymal fat. ND, normal diet (AIN-76); HFD, high-fat diet (45% kcal from fat); CLE, HFD+chrysanthemum leaf ethanol extract (1.5%, *w*/*w*); LU, HFD+luteolin (0.003% *w*/*w*). eWAT, epididymal white adipose tissue; UCP1, uncoupling protein-1; BAT, brown adipose tissue.

**Figure 3 nutrients-11-01347-f003:**
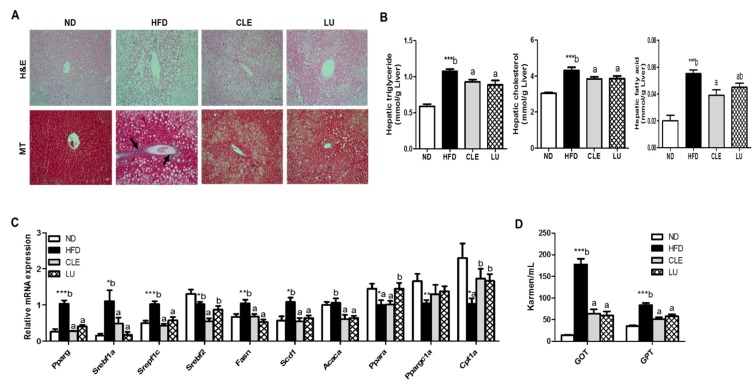
Effect of chrysanthemum leaf ethanol extract and luteolin supplements for 16 weeks on the hepatic steatosis-related markers in C57BL/6J mice fed a high-fat diet. (**A**) morphology and masson’s trichrome stain in liver (magnification ×200); (**B**) hepatic lipid levels (**C**) expression of hepatic lipid-regulating genes (**D**) hepatotoxicity index. Data are mean ± SE. Significant differences between HFD compared to ND are indicated * *p* < 0.05, ** *p* < 0.01, and *** *p* < 0.001 (Student’s *t*-test). ^ab^ Means not sharing a common letter are significantly different among the HFD-fed groups at *p* < 0.05. ND, normal diet (AIN-76); HFD, high-fat diet (45% kcal from fat); CLE, HFD+chrysanthemum leaf ethanol extract (1.5%, *w*/*w*); LU, HFD+luteolin (0.003% *w*/*w*); H&E, hematoxylin and eosin; MT, masson’s trichrome; GOT, glutamic oxalacetic transaminase; GPT, glutamic pyruvic transaminase.

**Figure 4 nutrients-11-01347-f004:**
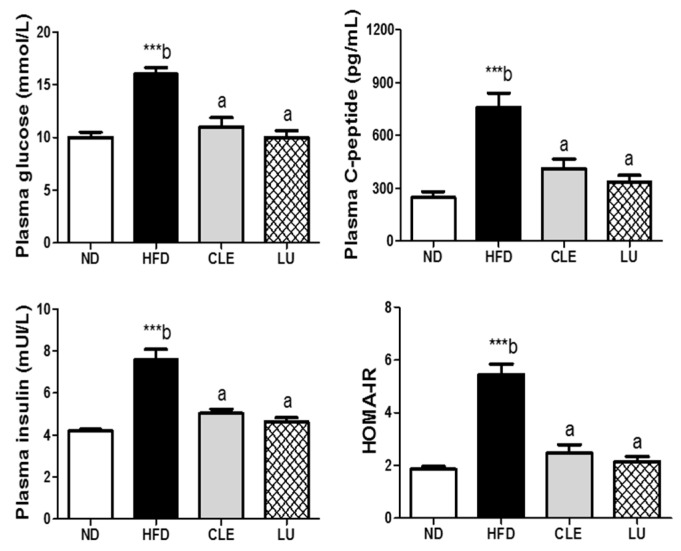
Effects of chrysanthemum leaf ethanol extract and luteolin supplements for 16 weeks on plasma glucose, insulin, C-peptide, and HOMA-IR in C57BL/6J mice fed a high-fat diet. Data are mean ± SE. Significant differences between HFD compared to ND are indicated *** *p* < 0.001 (Student’s *t*-test). ^ab^ Means not sharing a common letter are significantly different among the HFD-fed groups at *p* < 0.05. ND, normal diet (AIN-76); HFD, high-fat diet (45% kcal from fat); CLE, HFD+chrysanthemum leaf ethanol extract (1.5%, *w*/*w*); LU, HFD+luteolin (0.003% *w*/*w*); HOMA-IR, homeostasis model assessment of insulin resistance.

**Figure 5 nutrients-11-01347-f005:**
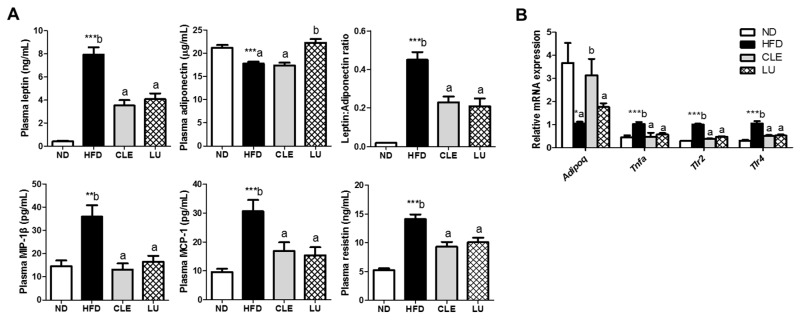
Effects of chrysanthemum leaf ethanol extract and luteolin supplements for 16 weeks on inflammation-related markers in C57BL/6J mice fed a high-fat diet. (**A**) Plasma adipokine and cytokine levels; (**B**) gene expression of adipokine and toll-like receptor genes in epididymal white adipose tissue. Data are mean ± SE. Significant differences between HFD compared to ND are indicated * *p* < 0.05, ** *p* < 0.01, *** *p* < 0.001 (Student’s *t*-test). ^ab^ Means not sharing a common letter are significantly different among the HFD-fed groups at *p* < 0.05. ND, normal diet (AIN-76); HFD, high-fat diet (45% kcal from fat); CLE, HFD+chrysanthemum leaf ethanol extract (1.5%, *w*/*w*); LU, HFD+luteolin (0.003% *w*/*w*); MIP-1β, macrophage inflammatory protein 1 beta; IL, interleukin; MCP-1, monocyte chemotactic protein 1; Adipoq, adiponectin; Tnfa, tumor necrosis factor α; Tlr, toll-like receptor.

**Figure 6 nutrients-11-01347-f006:**
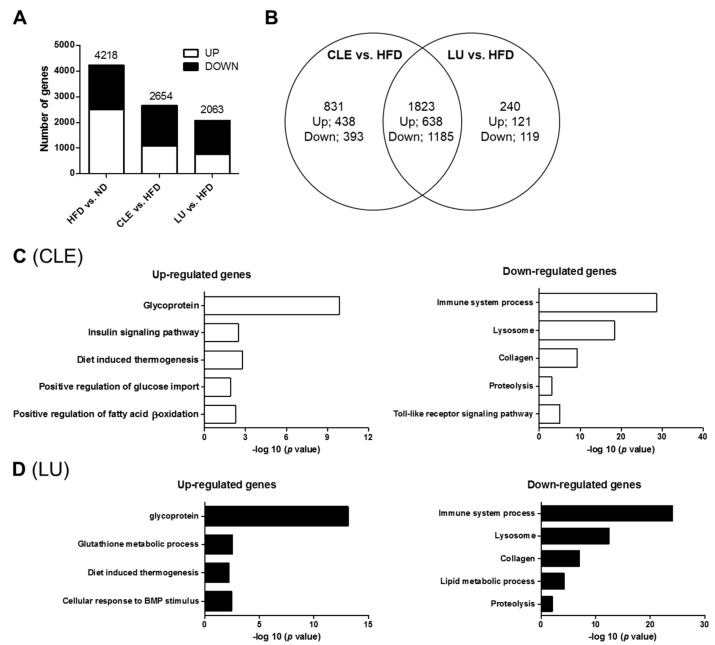
Transcriptional changes in eWAT of chrysanthemum leaf ethanol extract and luteolin groups compared to the high-fat diet group. (**A**) The numbers of up- or down-regulated genes in eWAT; (**B**) analysis of gene ontology in eWAT. Differentially expressed genes based on HFD compared to ND, CLE and LU compared to HFD comparison at *q* < 0.05, fold change > 1.5; (**C**,**D**) functional annotation clustering of gene ontology in eWAT in the CLE group and LU group, respectively. ND, normal diet (AIN-76); HFD, high-fat diet (45% kcal from fat); CLE, HFD+chrysanthemum leaf ethanol extract (1.5%, *w*/*w*); LU, HFD+luteolin (0.003% *w*/*w*, eWAT, epididymal adipose tissue).

**Figure 7 nutrients-11-01347-f007:**
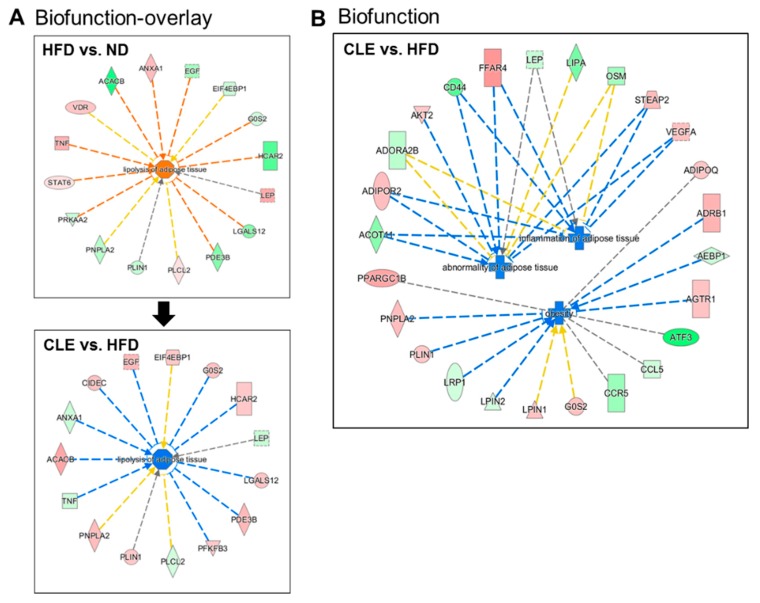
Biofunction analysis of differentially expressed genes from eWAT (**A**) Biofunction-overlay of the disease and function related to lipolysis in eWAT; (**B**) the disease and function in response to CLE in eWAT. Differentially expressed genes based on HFD compared to ND, and CLE and LU compared to HFD comparison for *q*- < 0.05, fold change > 1.5. ND, normal diet (AIN-76); HFD, high-fat diet (45% kcal from fat); CLE, HFD+chrysanthemum leaf ethanol extract (1.5%, *w*/*w*); LU, HFD+luteolin (0.003% *w*/*w*); eWAT, epididymal adipose tissue. Red indicates upregulation relative to HFD; green indicates downregulation relative to HFD. Orange color indicates activation of biofunction; blue color indicates inhibition of biofunction. Grey nodes indicate input genes into the pathway analysis and the different symbols indicate gene functions. Yellow nodes indicate inconsistent effect with predicted. Cross shape, disease; octagon shape, function; square, cytokine; triangle, phosphatase; horizontal oval, transcription regulator; vertical diamond; enzyme; circle, other.

**Table 1 nutrients-11-01347-t001:** Effects of chrysanthemum leaf ethanol extract and luteolin supplements for 16 weeks on plasma lipid levels in C57BL/6J mice fed high-fat diet.

Constituent	ND	HFD	CLE	LU
FFAs (mmol/L)	0.69 ± 0.03	0.82 ± 0.06 *	0.75 ± 0.04 ^#^	0.76 ± 0.05
TGs (mmol/L)	0.82 ± 0.06	0.98 ± 0.02 *	1.05 ± 0.05	1.05 ± 0.06
PLs (mg/dL)	276.74 ± 9.61	376.23 ± 16.90 ***^b^	329.75 ± 11.72 ^a^	327.28 ± 12.71 ^a^
Total-C (mmol/L)	3.95 ± 0.19	7.01 ± 0.38 ***^b^	5.59 ± 0.25 ^a^	5.40 ± 0.25 ^a^
HDL-C (mmol/L)	1.56 ± 0.04	1.88 ± 0.04 ***^a^	2.08 ± 0.05 ^b^	1.82 ± 0.05 ^a^
Non-HDL-C (mmol/L)	2.54 ± 0.15	5.48 ± 0.50 ***^b^	3.75 ± 0.25 ^a^	4.23 ± 0.23 ^a^
HTR	39.90 ± 1.15	26.87 ± 1.60 ***^a^	36.94 ± 2.11 ^b^	34.87 ± 1.78 ^b^
Apo A1 (mg/dL)	20.24 ± 0.27	19.18 ± 0.18 **^a^	20.85 ± 0.20 ^b^	20.43 ± 0.27 ^b^
Apo B (mg/dL)	3.61 ± 0.76	5.88 ± 0.51 ***^b^	3.71 ± 0.42 ^a^	4.08 ± 0.40 ^a^

Data are mean ± SE. Significant differences between HFD group compared to ND group are indicated at * *p* < 0.05, ** *p* < 0.01, *** *p* < 0.001, and CLE group compared to HFD group are indicated # *p* < 0.05 (Student’s *t*-test). ^ab^ Means not sharing a common letter are significantly different among the HFD-fed groups at *p* < 0.05. ND, normal diet (AIN-76); HFD, high-fat diet (45% kcal from fat); CLE, HFD+chrysanthemum leaf ethanol extract (1.5%, *w*/*w*); LU, HFD+luteolin (0.003% *w*/*w*); FFAs, free fatty acids; TGs, triglycerides; PLs, phospholipids; total-C, total cholesterol; HDL-C, high-density lipoprotein cholesterol; non-HDL-C = (total-C) − (HDL-C); HTR, (HDL-C/total-C) × 100; Apo A1, apolipoprotein A1; Apo B, apolipoprotein B.

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
