# Peer review of "Chrysanthemum Leaf Ethanol Extract Prevents Obesity and Metabolic Disease in Diet-Induced Obese Mice via Lipid Mobilization in White Adipose Tissue"

_nutrients, 2019, doi:10.3390/nu11061347_

Round 1
Reviewer 1 Report
The author investigated the effect of chrysanthemum leaf ethanol extract on prevention of obesity and metabolic disease using obese mice. The experimental procedure are valid and adequate and the proposed molecular mechanisms are reliable in its quality and quantity.
Major points
1) The reviewer could not understand biofunction analysis chart in Fig. 7A and 7B. Are there any difference between the type of symbols (circle, diamond, triangle, square) of gene name? Are there any difference between circle and cross symbol? Please describe in legend.
2) Please show respiratory quotient in light and dark in Figure 2.
3) Line 197-200 and Fig. 2C. Please show the sufficient evidence that these observations are correct results.
Minor points
Line 83: Please show the basis of the amount of CLE (1.5%) and LU (0.003%) and the difference between CLE and LU used in experimental diet.
Line 84: Please show the nutritional composition of four diets as supplemantal table.
Line 320: The author do not consider "CLE inhibited lipolysis" from your experimental data.
Line 339: Please check the expression of "HFD-fed mice reduced the TG content" and compare the data shown in Table 1.
Line 340. Please check the expression of "increased the FFA" and compare the data shown in Table 1.
Author Response
Reviewer 1
Comments:
he author investigated the effect of chrysanthemum leaf ethanol extract on prevention of obesity and metabolic disease using obese mice. The experimental procedure are valid and adequate and the proposed molecular mechanisms are reliable in its quality and quantity.
Major points
1) The reviewer could not understand biofunction analysis chart in Fig. 7A and 7B. Are there any difference between the type of symbols (circle, diamond, triangle, square) of gene name? Are there any difference between circle and cross symbol? Please describe in legend.
è Thank you for your kind comment. We added legend about biofunction analysis with IPA.
Results (page 11, lines 345-350)
“Red indicates up-regulation relative to HFD; green indicates down-regulation relative to HFD. Orange color indicates activation of biofunction; blue color indicates inhibition of biofunction. Grey nodes indicate input genes into the pathway analysis and the different symbols indicate gene functions. Yellow nodes indicate inconsistent effect with predicted. Cross shape, disease; octagon shape, function; square, cytokine; triangle, phosphatase; horizontal oval, transcription regulator; vertical diamond; enzyme; circle, other.”
2) Please show respiratory quotient in light and dark in Figure 2.
è According to your comments, we add the respiratory quotient data to Figure 2B. Also, we revised several sentences and added on setence in methods and results.
Material and Methods (page 3, lines 97-98)
“The respiratory quotient (RQ) is automatically calculated as the ratio VCO2 and VO2.The EE was calculated as follows”
Results (page 5, lines 202-204)
“ND group exhibited a high RQ compared to the HFD-fed groups. RQ was lower during the dark cycle in CLE and LU groups than in the HFD group, suggesting that CLE might utilize carbohydrate in HFD-fed mice.”
3) Line 197-200 and Fig. 2C. Please show the sufficient evidence that these observations are correct results.
è Excess visceral WAT induces more activated liberation of FFA by lipolysis and is closely correlated with metabolic abnormalities.
To investigate the effect on lipolysis of HFD and CLE, we measured the concentrations of triglyceride and fatty acid in epididymal white adipose tissue (eWAT) in direct by comparing their concentrations between per gram of eWAT and total eWAT to exact explanation (Figure 2C). Since HFD group showed increased weight of eWAT, we needed to adjust the lipid concentrations applied with per g eWAT. Altered adipose tissue lipolysis means that HFD, CLE and LU might regulate adipocyte lipolysis
As results, HFD increased FA per g eWAT, but decreased TG per g eWAT compared to the ND group. However, TG concentration of HFD group was markedly elevated because of increased eWAT weight.
In CLE and LU groups, there were no significant difference in adipose triglyceride level (mg/g eWAT) compared to the HFD group, while led to a significant reduction in TG level in total eWAT because of decreased eWAT weight. However, CLE and LU significantly reduced both the concentrations of FA/g and FA in total eWAT than HFD group. Consistent with the adipose fatty acid, CLE and LU supplement significantly lowered plasma FFA level compared to the HFD group.
Moreover, data (gene expression, hepatic lipid level) could be sufficient evidence consistent with lipid levels of adipose tissue, representing our data is correct results.
Together, Figure 2C data indicates that CLE and LU regulate adipocyte lipolysis, thereby resulting in reduction of plasma FFA, FFA influx and hepatic steatosis.
è Also, we revised several sentences.
Results (page 5, lines 204-210)
“Notably, TG concentration per gram of eWAT was significantly lowered in the HFD group than in the ND group, CLE and LU supplement did not affect it. However, when TG concentration converted in total e WAT, the result is reversed by showing markedly elevated compared to the ND group due to increased weight of eWAT in HFD group. In addition, fatty acid (FA) concentration was significantly higher in the HFD group than in the ND group. However, CLE and LU supplements significantly reduced both the concentrations of FA/g and FA in total eWAT than HFD-fed mice (Figure 2C), consistent with reduced plasma FFA level.”
Minor points
Line 320: The author do not consider "CLE inhibited lipolysis" from your experimental data.
è High rate of lipolysis of the visceral white adipose tissue induces delivery of free fatty acid to the liver via the portal vein, leading to hepatic steatosis and lipotoxicity.
In the present study, HFD increased fatty acid levels in both of plasma and epididymal adipose tissue (eWAT), and lipolysis associated genes expression in eWAT, while HFD down regulated lipogenesis associated gene, thus representing HFD activated lipolysis (Table 1, Figure 2C, 2E, Figure 7A). However, CLE down-regulated the lipolysis associated genes, and up-regulated lipogenic genes in eWAT. In addition, CLE supplement reduced both FA in plasma and liver (Table 1, Figure 3B).
‘CLE inhibited lipolysis’ is supported following experimental data;
Adipose lipid level (Figure 2C) / plasma FFA (Table 1) / hepatic lipid level (Figure 3B), Adipocyte gene expression (Figure 2E).
è To avoid confusion, we revised several sentences from ‘CLE inhibited lipolysis’ to ‘CLE inhibited excess lipolysis’.
Line 339: Please check the expression of "HFD-fed mice reduced the TG content" and compare the data shown in Table 1.
Line 340. Please check the expression of "increased the FFA" and compare the data shown in Table 1.
è HFD-fed mice reduced the TG/g eWAT (Figure 2C, upperpart left) and significantly increased the plasma FFA level (Table 1) those than in the ND group, although plasma TG level was not altered.
è Thank your comment. We checked the expression and revised sentence.
Discussion (page 11, lines 352-354)
“HFD-fed mice reduced the TG/g content in eWAT and increased the FA concentration in both plasma and eWAT compared with ND-fed mice, indicating HFD induced and regulated adipose lipolysis.”
Line 83: Please show the basis of the amount of CLE (1.5%) and LU (0.003%) and the difference between CLE and LU used in experimental diet.
è Thank you for your comment why we chosen the doses of CLE and LU. First, in previous study, the dose of chrysanthemum flower extract was 1% and 5 % (5 g/100 g diet, w/w), however only 5% (w/w) dose showed anti-diabetic effect in vivo (Yamamoto J et al., 2013). Second, there were beneficial effect on obesity and metabolic disease at luteolin 0.005% (w/w) (Kwon EY et al., 2014).
Therefore, in the present study, 0.003% of luteolin and 1.5 % of CLE were decided for in investigating the biological effects at the lower dose than 0.005% luteolin and 5% chrysanthemum extract, respectively.
è Also, we add several sentence.
Material and Methods (page 2, lines 84-87)
“The dose of CLE and luteolin was based on previous in vivo studies, 0.005% (w/w) luteolin and 5% (w/w) chrysanthemum flower extract, respectively [15, 17]. 0.003% (w/w) luteolin and 1.5% (w/w) CLE were decided for in investigating the biological effects at the lower dose than doses in previous study.”
Yamamoto, J.; Tadaishi, M.; Yamane, T.; Oishi, Y.; Shimizu, M.; Kobayashi-Hattori, K. Hot water extracts of edible Chrysanthemum morifolium Ramat. exert antidiabetic effects in obese diabetic KK-Ay mice. Biosci Biotechnol Biochem 2015, 79, 1147-1154,
Kwon, E.-Y.; Jung, U.J.; Park, T.; Yun, J.W.; Choi, M.-S. Luteolin attenuates hepatic steatosis and insulin resistance through the interplay between the liver and adipose tissue in diet-induced obese mice. Diabetes 2015, 64, 1658-1669, doi: 10.2337/db14-0631.
Line 84: Please show the nutritional composition of four diets as supplemantal table.
è According to your suggestions, we added diet composition for animal experiment in supplemental table (Table S2).

Reviewer 2 Report
Very exhaustive work about the effects of supplements of chrysanthemum leaf extracts on the fattening capacity of a high-lipid diet in obese mice. The effects of the high-lipid diet supplemented with an alcoholic extract (CLE) of said leaf or with the direct supplement of one of the flavonoids it contains, luteolin (LU), have been proven. It has been proven that both supplements revert (to a large extent) the increase in weight, especially of adipose tissue, while decreasing the presence of liver lipids. The extracts also reverse the modifications on the expression of various genes related to the lipid and energy metabolism of adipocytes and hepatocytes. The treatment with extracts also decreases the increases caused by the high lipid diet in different plasma markers such as insulin, leptin and hepatic transaminases. The effects of the treatment with these extracts go much further, since several hundred genes can be affected, both up regulated and down regulated.
The main conclusion of the authors is that these supplements can improve obesity and its comorbidities. In fact, the results obtained support this interpretation.
The experiment is comprehensive and well planned. The techniques used seem the most appropriate and the results seem clear.
Despite this, there are some aspects that the authors should clarify:
When they talk about food efficiency ratio (ln 176) they should define it more clearly, since it is supposed to be the result of dividing body weight gain (g / per week) by food intake (g / week). Therefore the units must be g of weight gain / g food, units that do not appear in Figure 1 B. If this is so, it would not be better to express the relationship with reference to the energy ingested, instead of the weight of ingested feed ?, ie. Weight gain (g)/energy ingested (kJ)
Therefore, it would be better if the data of Figure 1B referring to the intake of grams of feed, were changed by kJ of energy ingested.
The first paragraph of the discussion is an almost literal repetition of a fragment of the introduction. Therefore, it should be deleted.
The authors claim that the size of the adipocytes shown in Figure 2A is greater in the animals treated with the high lipid diet. Although they refer that the pictures are made at the same magnification scale, it would be better if they were accompanied by measurements in micrometers.
The effects of CLE and LU are very similar, since there are hardly significant differences between their effects when we contemplate the results of hormone levels, gene expression, etc. However, if we look at the relationship between genes that change their expression when comparing CLE / HFD (438 up and 393 down) and LU / HFD (121 up and 119 down), it seems evident that the effects of CLE seem more potent than those of LU. In addition, taking into account that as shown in Table S1, the LU content in the CLE extract is very small (0.008%) compared to the total flavonoid content (0.195%), so that only 4% of the content of CLE, seems to indicate, that at least with regard to genes that regulate lipid metabolism and inflammation factors, both in adipose tissue and liver, the role of LU could potentially be more powerful than that of the rest of flavonoids. Do not the authors believe that they should make a consideration about it in the discussion?
Author Response
Despite this, there are some aspects that the authors should clarify:
1) When they talk about food efficiency ratio (ln 176) they should define it more clearly, since it is supposed to be the result of dividing body weight gain (g / per week) by food intake (g / week). Therefore the units must be g of weight gain / g food, units that do not appear in Figure 1 B. If this is so, it would not be better to express the relationship with reference to the energy ingested, instead of the weight of ingested feed?, ie. Weight gain (g)/energy ingested (kJ)
Therefore, it would be better if the data of Figure 1B referring to the intake of grams of feed, were changed by kJ of energy ingested.
è Thank you for your important comment. According to your suggestion, we represent energy intake data instead of food efficiency ratio (Figure 1B).
2) The first paragraph of the discussion is an almost literal repetition of a fragment of the introduction. Therefore, it should be deleted.
è As your suggestion, we deleted the paragraph, also revised reference.
3) The authors claim that the size of the adipocytes shown in Figure 2A is greater in the animals treated with the high lipid diet. Although they refer that the pictures are made at the same magnification scale, it would be better if they were accompanied by measurements in micrometers.
è Thank you for your comments. It is our mistake. Actually, we observed morphology and size of adipocyte at magnification scale × 200, however we didn’t put the scale bar. Instead, we provided a description the magnification scale in material and methods and figure legend.
4) The effects of CLE and LU are very similar, since there are hardly significant differences between their effects when we contemplate the results of hormone levels, gene expression, etc. However, if we look at the relationship between genes that change their expression when comparing CLE / HFD (438 up and 393 down) and LU / HFD (121 up and 119 down), it seems evident that the effects of CLE seem more potent than those of LU. In addition, taking into account that as shown in Table S1, the LU content in the CLE extract is very small (0.008%) compared to the total flavonoid content (0.195%), so that only 4% of the content of CLE, seems to indicate, that at least with regard to genes that regulate lipid metabolism and inflammation factors, both in adipose tissue and liver, the role of LU could potentially be more powerful than that of the rest of flavonoids. Do not the authors believe that they should make a consideration about it in the discussion?
è Thank you for your raising an important comment. Our study was carried out to clarify that multi-compounds synergistically affects obesity and its comorbidities in diet-induced obese mice with positive control, luteolin which powerful bioactive compound. As referred in materials, CLE contained luteolin 0.008%. However, we hypothesized that CLE has synergy effect in combination luteolin, its glycoside and chlorogenic acid. In CLE, luteolin-7-O-β-D-glucuronide showed antioxidative as well as AGE-inhibitory activities. Chlorogenic acid is well known to have anti-obesity and anti-diabetic effects.
As you mentioned, CLE supplement changed biochemical profiles and gene expression pattern in similar with LU-fed mice. However, CLE/HFD regulated a various the gene expression more than LU/HFD, although a total of 1823 common genes were identified among the CLE and LU groups. In functional annotation clustering data (Figure 6C, D), common GO term is thermogenesis, lipid metabolism and immune system process, while insulin signaling of GO term is enriched by CLE supplement.
This suggests a synergistic effect of the bioactive phytochemical compounds in CLE, but luteolin seemed to be more potent for obesity and related metabolic disorders than other compounds.
Thus, these results indicate that luteolin in CLE may be responsible for its bioactive compound.
Results (page 13, lines 441-450; 321-325)
“In addition, a total of 1823 common genes were identified among the CLE and LU groups (Figure 6B). However, CLE supplement only 831 DEGs were regulated, and LU supplement only 240 DEGs were upregulated in among the high-fat diet-responsive genes (Figure 6B).”
“Among the 1304 genes down-regulated by LU supplementation, the immune system process, lysosome, collagen, the lipid metabolic process, and proteolysis were enriched GO terms. Moreover, CLE and LU groups have common GO terms which is thermogenesis, lipid metabolism and immune system process, while insulin signaling GO term is enriched by CLE supplement.”
Discussion (page 13, lines 441-450)
“In the present study, the CLE supplement changed biochemical profiles and gene expression pattern in similar with LU group. However, CLE/HFD regulated a various the gene expression more than LU/HFD in mRNA sequencing analysis, although a total of 1823 common genes were identified among the CLE and LU groups. CLE contained luteolin 0.008%, only 4 % of CLE. It is supposed that CLE has synergy effect in combination luteolin, its glycoside and chlorogenic acid. Among flavonoids present in CLE, luteolin-7-O-β-D-glucuronide exhibited antioxidative as well as AGE-inhibitory activities. Chlorogenic acid is well known to have anti-obesity and anti-diabetic effects. This suggests a synergistic effect of the bioactive phytochemical compounds in CLE, but luteolin seemed to be more potent for obesity and related metabolic disorders than other compounds. Thus, these results indicate that luteolin in CLE may be responsible for its bioactive compound.”

Round 2
Reviewer 1 Report
The author improved the manuscript according to the reviewer's comments.
Reviewer 2 Report
The authors have solved the questions raised by referees.